# Comparative Pre-Clinical Analysis of CD20-Specific CAR T Cells Encompassing 1F5-, Leu16-, and 2F2-Based Antigen-Recognition Moieties

**DOI:** 10.3390/ijms24043698

**Published:** 2023-02-12

**Authors:** Tatyana Belovezhets, Sergey Kulemzin, Olga Volkova, Alexander Najakshin, Alexander Taranin, Andrey Gorchakov

**Affiliations:** 1Almazov National Medical Research Centre, 197341 Saint Petersburg, Russia; 2Institute of Molecular and Cellular Biology of the SB RAS, 630090 Novosibirsk, Russia

**Keywords:** CAR T cell, ALL, NHL, CD20, cancer immunotherapy

## Abstract

Over the past decade, CAR T cell therapy for patients with B cell malignancies has evolved from an experimental technique to a clinically feasible option. To date, four CAR T cell products specific for a B cell surface marker, CD19, have been approved by the FDA. Despite the spectacular rates of complete remission in r/r ALL and NHL patients, a significant proportion of patients still relapse, frequently with the CD19 low/negative tumor phenotype. To address this issue, additional B cell surface molecules such as CD20 were proposed as targets for CAR T cells. Here, we performed a side-by-side comparison of the activity of CD20-specific CAR T cells based on the antigen-recognition modules derived from the murine antibodies, 1F5 and Leu16, and from the human antibody, 2F2. Whereas CD20-specific CAR T cells differed from CD19-specific CAR T cells in terms of subpopulation composition and cytokine secretion, they displayed similar in vitro and in vivo potency.

## 1. Introduction

CAR T cell therapy for patients with B cell neoplasms has made rapid strides from an experimental approach into a potent treatment modality, with four CD19-specific CAR T cell products approved by the FDA to date. Importantly, 20–30% of r/r ALL patients who received CD19-targeted CAR T cells eventually progress with CD19-negative relapses [1,2,3,4,5], which prompts the development of “second-line” CAR T cells targeting additional B cell surface markers. Two such proteins, CD20 and CD22, appear to be the most promising as multiple CD20- and CD22-specific monoclonal antibodies and BiTEs with excellent safety and efficacy profiles have been used in the clinic.

Clinical trials of CD22-specific CAR T cells to treat r/r B-ALL patients, many of who have had a prior CD19-targeted therapy, have resulted in complete responses in 57% [6] to 70% [7] of patients. Notably, moderate cell surface density of CD22 was observed to further decrease during the CD22-targeted CAR T cell therapy and to result in CD22-negative/low relapses. To overcome the issue of low efficacy in patients with CD22-low blasts as well as the possible immunogenicity of murine scFv-derived CARs, it is reasonable to design fully human CD22-specific CARs derived from high affinity human antibodies. This should translate into improved outcomes in patients who have received prior CD22-targeting therapeutics.

Along with CD19, CD20 was historically one of the first protein targets for CAR T cell therapy [8,9] due to the prevalence of CD20-positive cancers among B cell neoplasms and the high surface density of CD20 on malignant B cells [10]. Several CD20-targeted CAR T cell products have been successfully tested in clinical trials for patients with B cell leukemia and several types of B cell lymphomas. These CAR designs were based on the scFvs derived from the monoclonal antibodies, 1F5 [11], Leu16 [9,12], and 2F2 [13]. Clinical and pre-clinical comparative analyses of these CAR designs and CAR T cell products were never performed due to the numerous differences in the CAR modules, manufacturing details, mouse models, and patient selection. This prompted us to (i) redesign a fully human CD20-specific CAR based on the sequence of the monoclonal antibody, 2F2 (ofatumumab) and (ii) perform accurate side-by-side in vitro and in vivo comparisons of three CD20-specific CARs that only differed in their antigen-recognition moieties (2F2 vs. 1F5 vs. Leu16).

## 2. Results

### 2.1. Engineering CD20-Specific CAR T Cells

In order to perform back-to-back comparisons of various CD20-specific CAR designs, we set out to engineer CARs that only differed in their antigen-recognition modules. To this end, three scFvs conferring CD20 specificity were selected, namely derived from the mouse monoclonal antibodies, 1F5 and Leu16, and from the human monoclonal antibody, 2F2 (ofatumumab) [14,15,16]. Appropriate sequences were produced by gene synthesis (Genewiz) and cloned into a backbone encoding the hinge and transmembrane regions of human CD8a, the 41BB-CD3z co-stimulation/activation domain, and an IRES-linked cell surface reporter, NGFR (Figure 1A). As a reference control, CD19-specific CAR based on the FMC63-derived scFv was used because it has been comprehensively characterized both pre-clinically and in the clinic (Kymriah CAR) [17,18,19]. Irrelevant PSMA-specific CAR based on the J591-derived scFv and employing the same backbone was used as a negative control as PSMA is absent on the surface of Nalm6-CD20 cells used as the targets.

To obtain CAR T cells and perform their side-by-side comparisons, primary human T cells were isolated from the PBMCs of four healthy men (median age, 30 years). CAR cassettes were delivered via lentiviral transduction. Average transduction levels, fold expansion on day 12, and % cytotoxicity in vitro across the CAR T cells obtained are summarized in Table 1. Following lentiviral transduction, the surface expression of all the CARs tested remained detectable, as assayed by indirect anti-NGFR staining and flow cytometry (Figure 1B) or by direct CAR detection using protein L staining (Appendix A). The signal/noise ratio was higher with the former approach and therefore, this CAR detection assay was selected for all of the downstream experiments.

### 2.2. The CAR T Cell Products Obtained Display a Similar Subpopulation Composition

Multiple lines of evidence indicate that the subpopulation composition of CAR T cell products may have a significant impact on their therapeutic activity [20,21,22,23,24,25]. This also applies to the CD4:CD8 CAR T cell ratio, with a ratio of ~1 showing the greatest in vivo activity [3,26]. We asked whether the panel of CD20-specific CAR T cells obtained would have distinct subpopulation makeups. To profile the T cell subsets, we used a FACS analysis where each of the CD4 or CD8 subpopulations was further subdivided into naive T cells (T_N_, CD45-RA+ CD62L+), central memory T cells (T_CM_, CD45-RA- CD62L+), effector memory T cells (T_EM_, CD45-RA- CD62L-), and terminally differentiated effector memory T cells (T_EMRA_, CD45-RA+ CD62L-). The flow cytometry primary data plots are given in the Appendix A. CAR T cells produced from healthy donors were phenotyped on day 3 following transduction with non-transduced T cells from the same donors serving as a reference control (Figure 2). The CD4:CD8 ratio varied in the range from 1.3 to 6.7 for the donors used in our work and was observed to decrease by day 21 (Figure 3). Long-term ex vivo cultivation of CAR T cell products obtained predictably resulted in their shifted subpopulation composition. Specifically, the T_EM_ and T_CM_ subpopulations as well as the CD8 CAR T cells in general are expanded (Figure 2), which is attributable to the presence of the exogenously supplemented cytokines, IL2 and IL15, in the culture medium. The subpopulation composition of the CAR T cells obtained did not show statistically significant differences.

Knowledge of the subpopulation makeup of CAR T cell products is typically complemented by the quantification of expression of exhaustion markers, PD-1, TIGIT, LAG3, and TIM3 [27,28]. Notably, although PD-1 expression on T cells has been strongly associated with a lack of therapeutic effects [29,30], it can also be a marker of recent T cell activation [31]. Therefore, the simultaneous expression of several exhaustion markers appears to be a more reliable marker of genuinely exhausted T cells. To characterize our CAR T cell products more comprehensively, the surface expression of CAR and three exhaustion markers–PD-1, LAG3, and TIM3–was assessed by flow cytometry using the cells collected on day 8 of the serial killing assay. Although approximately 50% of the CAR T cells were PD-1-positive, all three exhaustion markers were found in fewer than 20% of the CAR T cells–despite continuous stimulation with target cells and being grown in a poor culture medium in the absence of exogenous cytokines (Figure 4). Irrelevant (control) CAR T cells that did not receive strong activation signals from the target cells displayed nearly background levels of PD-1 and other exhaustion markers. This observation is consistent with the idea that this phenotype is largely due to antigen-specific and CAR-mediated stimulation of CAR T cells. No significant differences were observed across the CD20-specific CAR T cell products tested, nor did they differ significantly in terms of the activation level or percentage of exhausted CAR T cells from the reference FMC63-based CAR.

### 2.3. Temporal Dynamics of CAR T Cell Composition

During in vitro cultivation or prolonged expansion, CAR T cells may undergo significant shifts in their composition, both in terms of the percentage of CAR-expressing T cells and the phenotypic subpopulations [2,3]. To understand how CAR design may contribute to the composition of CAR T cell products, we measured the percentage of CAR-positive T cells at two timepoints, day 3 and day 21. We observed that, on average, fewer CAR T cells were present in the products on day 21 (modeling the long-term expansion situation) compared to day 3 (Figure 5), which could be attributable to pseudotransduction and/or scFv-driven CAR T cell exhaustion and death [32,33]. Our data indicate that, regardless of the scFv module used for constructing the CARs, the temporal dynamics of CAR T cell positivity across our products is generally the same and the CD20-specific CARs are indistinguishable from the CD19-specific FMC63-CAR in this assay.

Given that the percentage of CAR T cells in CAR T cell products is not the only parameter known to contribute to the efficacy and toxicity of the therapy, we asked whether there may be any changes in the subpopulation composition of our CAR T cell products, i.e., in the percentage of T_N_, T_CM_, T_EM_, and T_EMRA_ cells, on day 3 and day 21. To address this question, we performed immune phenotyping using appropriate antibodies and again observed no significant differences between CD4- or CD8-gated CAR T cell subpopulations (Figure 2A,B). Taken together, these data indicate that the exact choice of the scFv for CAR design had little, if any, influence on the phenotypic composition of the CD20-specific CAR T cell products obtained in the absence of stimulation with the cognate antigen. We then turned to the model of chronic antigen stimulation wherein CAR T cells are incubated with target cells over a period of eight days (see more details in the next section) and immunophenotyped. Under these conditions, profound subpopulation shifts and an increase in the percentage of T_CM_ cells in the product (Figure 2C) were observed. Notably, all CD20-based CAR T products were enriched with the unwanted T_EM_ cells, in contrast to the FMC63-CAR T product, which still displayed high T_CM_ content.

### 2.4. CD20-Specific CAR T Cells Display Similar Levels of Activation In Vitro

Successful CAR design critically depends on the choice of the antigen-recognition module in that it must provide antigen-specific activation of the T cell without an unwanted activation in the absence of the cognate antigen. Thus, CAR affinity and a wise choice of its Kon/Koff rates become central. Intuitively, the higher the affinity, the greater is the chance that such CAR T cells will recognize and destroy the targets displaying a low surface density of the antigen. On the other hand, the phenomenon of “affinity ceiling” is also known, i.e., increasing the affinity above a certain threshold may not necessarily result in increased cytotoxicity [34]. Thus, adjusting the CAR affinity or Kon/Koff rates may be required, but this in turn depends on the characteristic range of the antigen density on tumor cells and still largely remains a trial-and-error area of CAR design [35,36,37,38]. This is consistent with the observations that lower affinity of the CAR not only does not result in lower efficacy or persistence, but in fact may be a safer alternative as the risks of CRS are reduced [39].

To test how our CAR T cell products respond to cell targets displaying lower levels of CD20 expression, HEK293T cells were modified to ectopically express CD20 and two monoclones, F6 and B3, having lower CD20 expression compared to the CD19+ CD20+ lymphoma Raji cell line were obtained (Appendix A). In activation assays, CD19-specific FMC63-CAR T cells were only activated by Raji cells but not with any of the HEK293T-CD20 cell targets, as expected. All CD20-targeted CAR T cells displayed reduced expression of the early activation marker, CD69, upon co-incubation with CD20-low targets compared to their incubation with the positive control cells (Raji) (Figure 6). Notably, CAR T cell activation was reduced but not absent under these conditions. 1F5-bearing CAR T cells display significantly higher levels of activation upon stimulation with the lowest density clone, B3, when compared to the Leu16-based CAR T.

### 2.5. CD20-Specific CAR T Cells Display Similar “Instant” Cytotoxicity, Yet Distinct Sequential Killing Activity In Vitro

The ability of CAR T cells to selectively eliminate target cancer cells is one of the most important parameters assayed during CAR characterization. Here, we used Nalm6-CD20 cells pre-labeled with eFluor 670 dye to assess the “instant” cytotoxicity of various CAR T cells in a standard 4 h co-incubation assay (Figure 7).

No significant differences between the cytotoxic activities of CD20- and CD19-specific CARs were observed in this assay with all CARs behaving very similarly, which is consistent with the phenotyping data hereinabove. Whereas “instant” cytotoxicity is broadly accepted as a measure of correct target recognition, it may poorly model the situation in vivo when CAR T cells are facing the outnumbering tumor cells for a prolonged time. Therefore, we asked whether CAR T cells that are chronically stimulated with target cells might display any differences in cytotoxicity. Under these conditions, only irrelevant PSMA-specific CAR T cells were significantly different from the rest of the CAR T cells in terms of their tumor elimination potency across the time points explored. Yet, we noted that 1F5-CAR T cells displayed a non-significant trend toward losing their anti-tumor control on day 8 (Figure 8).

In order to characterize the in vitro cytotoxic properties of the CAR T cells obtained more comprehensively, the IL2 and IFNg levels in the supernatants of CAR T cells co-incubated with tumor cells for 12 h at a 1:2 ratio were measured by ELISA. As expected, all the CAR T cells displayed significantly greater IL-2 and IFNg secretion compared to the irrelevant PSMA-specific CAR T cells (J591-CAR) (Figure 9). Interestingly, both cytokines were secreted at significantly lower levels by the FMC63-CAR T cells compared to the CD20-specific CAR T cells. These data call for caution as the overactivation of CAR T cells (measured here indirectly by their cytokine secretion) is known to be associated with a life-threatening condition referred to as cytokine release syndrome, warranting further analysis in appropriate pre-clinical models [40,41,42,43,44,45,46].

### 2.6. CAR T Cells Display Distinct Cytotoxic Activity In Vivo

Pre-clinical analysis of the in vivo activity of CAR T cells relies on the immunodeficient mouse models xenotransplanted with human cancer cells. Multiple formats of the in vivo tests have been described in the literature, with a broad range of tumor/CAR T cell numbers used as well as various treatment regimens [39,47]. We chose to test our CAR T cells in a format wherein experimental animals carry well-established tumors and CAR T cells are infused so that 100% of the tumor-engrafted animals are ultimately dead in the positive-control (FMC63-CAR) group. Specifically, NSG mice were engrafted with 3 × 10^5^ tumor cells and then treated with two infusions of CAR T cells (1 × 10^6^ cells (d8) and 2 × 10^6^ cells (d14)) (Figure 10A). Survival of the treated mice was used as the integral parameter informing CAR T cell efficacy. Median survival of the animals that received CD20- or CD19-CAR T cells was significantly different from that of the J591-CAR T-treated mice (Figure 10B, *p*-value < 0.001).

Our in vivo experiment involved a large tumor burden model and a delayed treatment scheme. No mice treated with 2F2- and FMC63-CAR T cells survived beyond day 80. Nonetheless, two animals in the Leu16- and one animal in the 1F5-CAR T cell group were observed to control the tumor and remained alive and overall healthy by day 108 until the experiment was terminated (Figure 10). The dynamics of tumor and CAR T cell numbers in the peripheral blood of experimental animals was monitored using qPCR. Bone marrow samples from the animals that had to be euthanized were also collected and analyzed for the presence of tumor and CAR T cells by FACS. Our analyses show that CAR T cell expansion occurred in all of the experimental groups and CAR T cell numbers in peripheral blood remained essentially the same until the mice were sacrificed (Appendix A). Notably, a peak of CAR T cell expansion around days 60–80 was observed in the Leu16-CAR T cell group, which matched the decline in tumor cell numbers (Appendix A). Bone marrow samples from the euthanized animals displayed abundant CAR T and tumor cells (Appendix A).

## 3. Discussion

Despite the significant progress of the CAR T cell field, one of the central questions in CAR design is how the inclusion of a specific CAR module will affect the multitude of CAR T cell features, such as cytotoxicity, proliferation, exhaustion during ex vivo cultivation and expansion, subpopulation composition, and persistence and efficacy in animal models and patients. In order to minimize the possible contribution of other CAR modules and focus on the antigen-recognition modules, three CD20-specific CARs were designed that differed only in the scFvs used. Specifically, we chose the previously reported and clinically tested 1F5 and Leu16 as well as the novel 2F2 (ofatumumab)-based antigen-recognition moiety. Following the transduction of T cells of four donors, CAR T cell products were successfully obtained and extensively characterized in a panel of in vitro tests and in vivo. “Gold standard” FMC63-based CAR of identical architecture, which is used in the Kymriah CAR T cell product, was used as a reference. We observed that in the 4 h “instant” cytotoxicity assay against Nalm6-CD20 target cells as well as in the sequential killing assay, CD20- and CD19-specific CARs behave very similarly. CD20-specific CAR T cells secrete significantly more IL-2 and IFN-g compared with the CD19-specific counterparts. Such a difference in cytokine secretion could be explained by the abundance of T_EM_ cells in CD20-specific CAR T products, as these cells are known to secrete more cytokines than T_CM_ cells [48]. These differences were not recapitulated in vivo, with CD19- and CD20-specific CAR T cells providing comparable levels of antitumor activity.

Clinical data are available for CD20-specific CAR T cell products based on the 1F5, Leu16, and 2F2 (CAR066)–each having a peculiar modular architecture–in NHL patients. Their efficacies appear overall comparable and result in 50–60% complete responses [11,12,49]. However, these studies have significant differences in the patient characteristics, particularly in terms of previous therapies and rituximab treatment, and cannot be reliably compared. Notably, the use of rituximab was demonstrated to result in CD20-negative/low escape [50]. Therefore, this substantiates the design and use of a 2F2 (ofatumumab)-based CAR, given that ofatumumab is active against a fraction of rituximab-resistant tumors [51]. Although multiple mechanisms are plausible, this effect is likely attributable to the use of a non-overlapping epitope of 2F2 on CD20 compared to those of rituximab, Leu16, or 1F5 (Figure 11).

Importantly, compared to the rest of the CD20-specific CARs, the 2F2-CAR design offers yet another advantage, namely it is fully human in sequence, which may reduce the chance of CAR T cell-associated toxicities and increase the persistence/efficacy of the CAR T cell product. The efficacy of the reference FMC63-based CD19-specific CAR, which has an antigen-recognition module of a mouse origin, is not known to be affected by the human anti-mouse antibody (HAMA) response, yet this may occur in the context of other scFvs and include T cell responses to foreign peptides [53].

As noted hereinabove, all CAR constructs used in this work were built on exactly the same CAR backbone; therefore, any differences in the phenotypes and activity of the CAR T cells are attributable to the nature of the scFv module and the fine details of its interaction with the target. Taken together, our data indicate that optimal CD20-specific scFv could not be readily identified based exclusively on the in vitro experiment data and highlight the fact that the efficacy of CAR T cell therapy is largely dictated by the nature of its target, including cell density, affinity, proximity to the membrane, etc. Notwithstanding, our preclinical evaluation of the 2F2-based CD20-specific CAR T cells demonstrates efficacy in the murine model comparable to the gold standard FMC63-CAR and paves the way to the clinical trial utilizing this product.

## 4. Materials and Methods

### 4.1. CAR Design

Lentiviral vector pCDH (System Biosciences, USA) was used as a backbone for CAR cloning. The nucleotide sequence of the reference FMC63-based CAR (Kymriah, Novartis, Cambridge, MA, USA) was obtained by gene synthesis (Genewiz, Plainfield, NJ, USA). It incorporated a unique AgeI (Thermo Fisher Scientific, Waltham, MA, USA) site immediately downstream of the mIgk-encoding sequence and a unique EcoRV (Thermo Fisher Scientific, Waltham, MA, USA) site in the middle of the CD8a hinge region. Next, AgeI/EcoRV-flanked cassettes encoding the scFv/partial CD8a hinge cassette corresponding to the CD20-specific antibodies, 1F5, Leu16, and 2F2, were similarly obtained by gene synthesis and pasted into the Kymriah CAR construct to substitute the FMC63-based antigen-binding module. J591-derived PSMA-specific scFv was cloned by PCR from a previously published construct [54,55,56] to obtain an irrelevant PSMA-specific CAR of an identical structural format. Therefore, all of the CAR constructs obtained had a well-described hinge, a transmembrane region derived from that of the human CD8a, and a second generation co-stimulation/signaling cassette encoding the fusion of the intracellular region of human 4-1BB and CD3ζ [57,58]. Finally, the expression of all the CAR constructs along with the IRES-linked NGFR surface marker was driven by the strong constitutive EF1a promoter. Lentiviral particles were obtained by co-transfection HEK293T cells with a combination of the CAR-encoding lentiviral construct of interest and the packaging plasmids, psPAX2 and pMD2.G, using a calcium phosphate transfection protocol as described [59]. Pseudovirus-containing supernatants were collected 48 h later, filtered through a PES filter (pore diameter 0.45 um) (TPP, Switzerland), and concentrated by centrifugation for 1.5 h at 35,000× *g*. Viral pellets were then resuspended in the T cell growth medium, snap-frozen in liquid nitrogen, and stored for up to 3 months at −70 °C.

Human B cell lymphoma cell line Nalm6 (ATCC, Manassas, VA, USA) was transduced with a lentiviral construct encoding human CD20 to obtain Nalm6-CD20 cells that stably expressed CD20.

### 4.2. CAR T Cell Production

Primary human peripheral blood mononuclear cells were isolated from a healthy human donor using ficoll (PanEco, Moscow, Russia) gradient centrifugation. T cells were then isolated and activated with CD3/CD28 Dynabeads (Thermo Fisher Scientific, USA) at a T cell:bead ratio of 1:1. Cells were cultured in a Tex-MACS (Miltenyi Biotec, Bergisch Gladbach, Germany) medium supplemented with 50 U/mL recombinant human IL-2 (Miltenyi Biotec, Bergisch Gladbach, Germany) and 7 ng/mL IL15 (Miltenyi Biotec, Bergisch Gladbach, Germany). T cells were transduced 24 h later using centrifugation at 600× *g* for 40 min at 32 °C in the presence of 10 mg/mL protamine sulfate and MOI = 3. Three days later, transduction efficiency was assessed using anti-NGFR (APC, clone ME20.4, BioLegend, San Diego, CA, USA) staining and flow cytometry.

### 4.3. Phenotypic Analysis of CAR T Cell Products by Flow Cytometry

FACS was used to measure CD4/CD8 ratios and enumerate CAR T cell subpopulations. CAR T cells were first gated by staining for NGFR (APC, clone ME20.4, BioLegend, USA). Next, CD4+ (FITC, clone LT4, Sorbent, Moscow, Russia) and CD8+ (APC-Fire750, clone HIT8a, BioLegend, San Diego, CA, USA) subpopulations were selected. Based on the expression of CD45-RA (PE, clone HI100, BioLegend, San Diego, CA, USA) and CD62L (PE-Cy7, clone DREG-56, BioLegend, San Diego, CA, USA), each of these subpopulations was further subdivided into naive T cells (T_N_, CD45-RA+ CD62L+), central memory T cells (T_CM_, CD45-RA- CD62L+), effector memory T cells (T_EM_, CD45-RA- CD62L-), and terminally differentiated effector memory T cells (T_EMRA_, CD45-RA+ CD62L-).

### 4.4. Phenotypic Analysis of Exhaustion of CAR T Cell Products by Flow Cytometry after Sequential Killing Assay

CAR T cells were mixed in the wells of a 96-well plate (TPP, Trasadingen, Switzerland) with Nalm6-CD20 at an E:T ratio of 1:2 (30,000 CAR T cells vs. 60,000 target cells per well) in IMDM (Gibco, Thermo Fisher Scientific, Waltham, MA, USA) medium with 10% FBS (Gibco, Thermo Fisher Scientific, Waltham, MA, USA) devoid of exogenous cytokines. Target cell and CAR T cell numbers in each well were quantified 48 h later using flow cytometry (unlike CAR T cells, target cells are CD19-positive). Then, fresh target cells were added to each well to reconstitute the initial 1:2 ratio. Cell counting and replating were repeated 4 times. After that, FACS was used to enumerate CAR T cell subpopulations. First, T cells were gated by staining for CD3 (clone HIT3, Alexa Fluor 488, BioLegend, San Diego, CA USA). CAR T cells were gated by staining for NGFR (Biotin, clone ME20.4, BioLegend, San Diego, CA, USA; Streptavidin, APC-Cy7, BioLegend, San Diego, CA, USA). Next, PD-1+ (PE/Cyanine7, clone EH12.2H7, BioLegend, San Diego, CA, USA), TIM3+ (APC, clone F38-2E2, BioLegend, San Diego, CA, USA), and LAG3+ (PE, clone 11C3C65, BioLegend, San Diego, CA, USA) subpopulations were selected.

### 4.5. Activation Assay

CAR T cells were mixed in the wells of a 96-well plate (TPP, Trasadingen, Switzerland) with Raji or monoclone HEK293T-CD20 with different levels of expression (Appendix A) at an E:T ratio of 1:2 (30,000 CAR T cells vs. 60,000 target cells per well) in IMDM (Gibco, Thermo Fisher Scientific, Waltham, MA, USA) medium with 10% FBS (Gibco, Thermo Fisher Scientific, Waltham, MA, USA) devoid of exogenous cytokines. Numbers of CD69+ CAR T cells in each well were quantified 4 h later using flow cytometry. CAR T cells were first gated by staining for CD3 (clone HIT3, Alexa Fluor 488, BioLegend, San Diego, CA, USA). Next, CD69+ (Biotin, clone FN50, BioLegend, San Diego, CA, USA; Streptavidin-PE, Invitrogen, Thermo Fisher Scientific, Waltham, MA, USA) subpopulations were selected.

### 4.6. In Vitro Cytotoxicity Assay

Nalm6-CD20 cells were labeled with eFluor670 (Thermo Fisher Scientific, Waltham, MA, USA) and transferred into a 96-well plate in Tex-MACS (Miltenyi Biotec, Bergisch Gladbach, Germany) medium without exogenous cytokine supplementation (60,000 cells/well). CAR T cells were then added to the Nalm6-CD20 targets in a cytokine-free Tex-MACS (Miltenyi Biotec, Bergisch Gladbach, Germany) medium at E:T ratios of 0.25, 0.5, 1.0, and 5.0. Four hours later, cell viability dye 7-AAD (Sigma, Burlington, MA, USA) was added to the wells and the percentage of surviving target cells was measured using flow cytometry. Cytotoxic activity (% of dead target cells) was calculated using the formula: (1 − live target cells/live target cells in the absence of CAR T cells) × 100%.

### 4.7. Sequential Killing Assay

CAR T cells were mixed in the wells of a 96-well plate (TPP, Trasadingen, Switzerland) with Nalm6-CD20+ cell targets at an E:T ratio of 1:2 (25,000 CAR T cells vs. 50,000 Nalm6-CD20 cells per well) in a Tex-MACS (Miltenyi Biotec, Bergisch Gladbach, Germany) medium devoid of exogenous cytokines. Target cell and CAR T cell numbers in each well were quantified 48 h later using flow cytometry (unlike CAR T cells, target cells are CD19-positive). Then, fresh target cells were added to each well to reconstitute the initial 1:2 ratio. Cell counting and replating were repeated 4 times.

### 4.8. ELISA

Prior to mixing with target cells (Nalm6-CD20), CAR T cells had their TexMACS medium replaced with IMDM. CAR-T cells were then co-cultured overnight with target cells at a 1:2 E:T ratio in IMDM medium containing 10% FBS without exogenous cytokines, in duplicate. The next day, cell cultures were centrifuged at 500× *g* and supernatants were transferred into new plates. Levels of human IFN-γ and IL-2 were analyzed by ELISA according to the manufacturer’s protocols (Vector Best, Russia).

### 4.9. CAR T Cell Activity In Vivo

To measure in vivo CAR T cell activity, female NSG mice (age 6–10 weeks) were used. On day 0, all of the mice received tail vein injections of 3 × 105 Nalm6-CD20 cells. On days 8 and 14, the mice were infused with 1 × 106 and 2 × 106 CAR T cells, respectively. The control mice received an irrelevant CAR T cell product directed against human PSMA (negative control). Tumor burden was assessed using two approaches, namely by qPCR analysis of gDNA isolated from serially collected blood samples and by qPCR and FACS analyses of endpoint samples (blood and bone marrow). Mice showing signs of terminal disease (leg paralysis, hunching) were euthanized and bone marrow/blood samples were collected.

### 4.10. qPCR

A GeneJET genomic DNA purification kit (Thermo Fisher Scientific, Waltham, MA, USA) was used to isolate genomic DNA from mouse blood and bone marrow samples. DNA was then used as a template for qPCR assays with the following primers and probe specific for the zeocin resistance gene (which was part of the CD20-expression cassette and was present in the Nalm6-CD20 targets): zeo_F 5′-AGTTGACCAGTGCCGTTCC-3′, zeo_R 5′-AAGTCGTCCTCCACGAAGTC-3′, zeo_P 5′-FAM-GGAGCGGTCGAGTTCTGGACCGAC-BHQ1-3′. To quantify the presence of CAR T cells, the primer/probe sequences were selected to anneal to the NGFR-encoding sequence: NGFR_F2 5′-GGCCTACATAGCCTTCAAGAGG-3′, NGFR_R2 5′-TCCACATAGCGTAAAAGGAGCAA-3′, NGFR_P2 5′-FAM-CAGCTAGTCGACAATCAACCTCTGGATTAC-BHQ1-3′. The following primer mix specific for the mouse Emid1 gene was used as an amplification control [60]: mEmid1_F 5′-GCCAGGACTGGGTAGCAC-3′, mEmid1_R 5′-AGGAGGCTCCTGAATTTGTGACAAG-3′, mEmid1_P 5′-FAM-CCTGGGTCATCTGAGCTGAGTCC-BHQ1-3′. All primers and probes were purchased from DNK-Sintez (Moscow, Russia).

### 4.11. Statistical Analyses

Analyses were performed using GraphPad Prism version 6 software. *p* values < 0.05 were considered significant. To analyze the possible differences in the subpopulation composition, proliferation rate, and cytotoxicity, multiple comparison tests from 2-way ANOVA were performed. For a comparison of the survival curves, a Log-rank (Mantel–Cox) test was used.

## Figures and Tables

**Figure 1 ijms-24-03698-f001:**
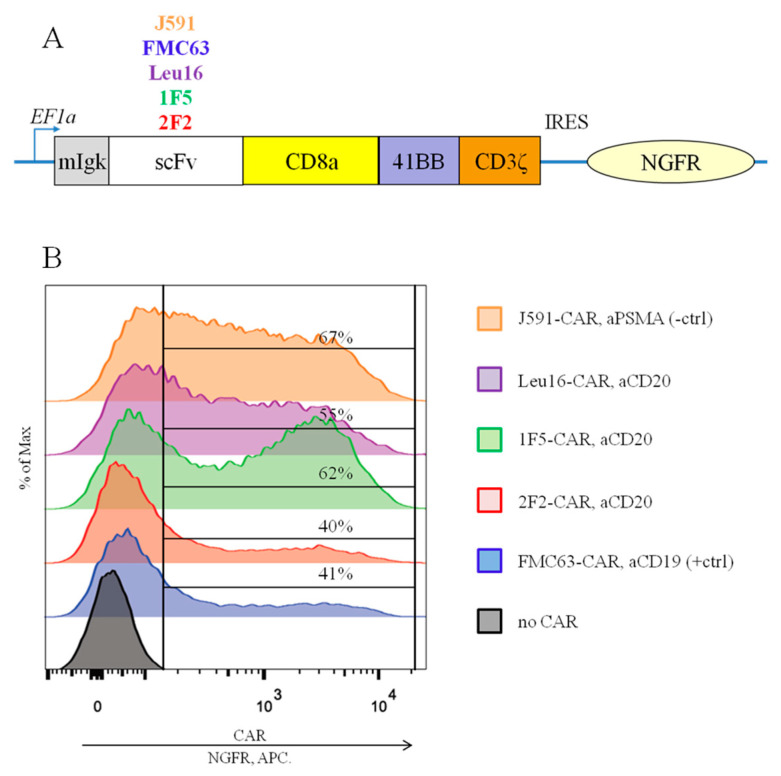
(**A**) Modular organization of the CARs being compared. (**B**) Similar surface expression of all the CARs tested was achieved following the transduction of primary human T cells, as assayed by anti-NGFR staining and flow cytometry (representative data for one of the donors are shown).

**Figure 2 ijms-24-03698-f002:**
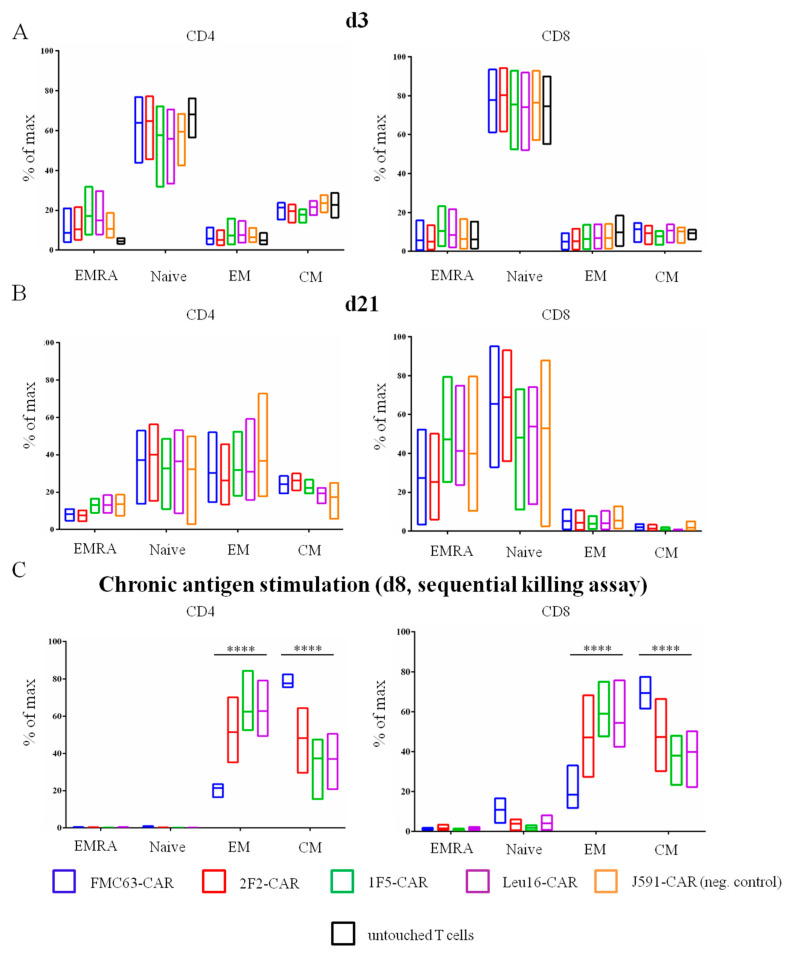
Percentage of T_N_, T_CM_, T_EM_, and T_EMRA_ cell subpopulations between CD4+ and CD8+ CAR T cells on (**A**) day 3 and (**B**) day 21 after isolation. “Untouched T cells” processed immediately after T cell isolation from blood were used as a reference. (**C**) Percentage of T_N_, T_CM_, T_EM_ and T_EMRA_ CAR T cell subpopulations on day 8 of the sequential killing assay. Mean values with range are shown, **** *p* < 0.0001.

**Figure 3 ijms-24-03698-f003:**
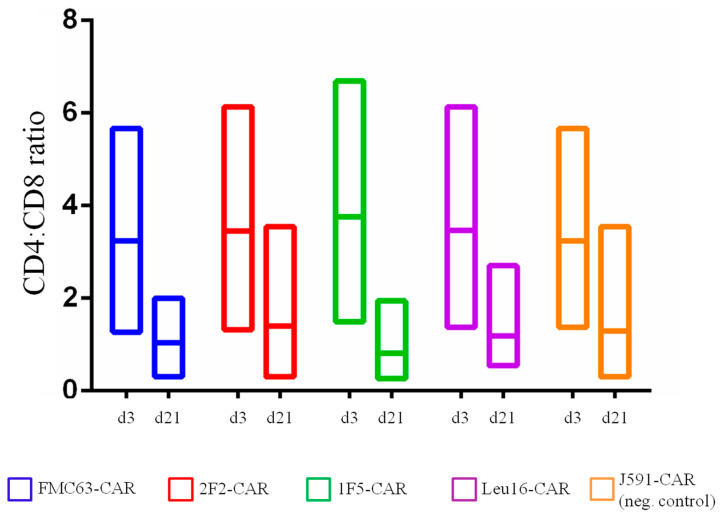
CD4:CD8 ratio in CAR T cells obtained on day 3 and day 21 of ex vivo cultivation (differences between day 3 and day 21 values are non-significant).

**Figure 4 ijms-24-03698-f004:**
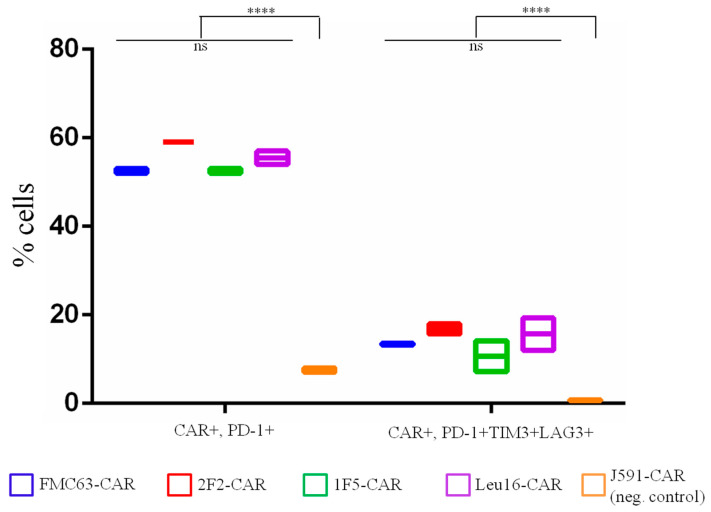
Percentage of exhausted CAR T cells on day 8 of the sequential killing assay. Mean values with range are shown, **** *p* < 0.0001.

**Figure 5 ijms-24-03698-f005:**
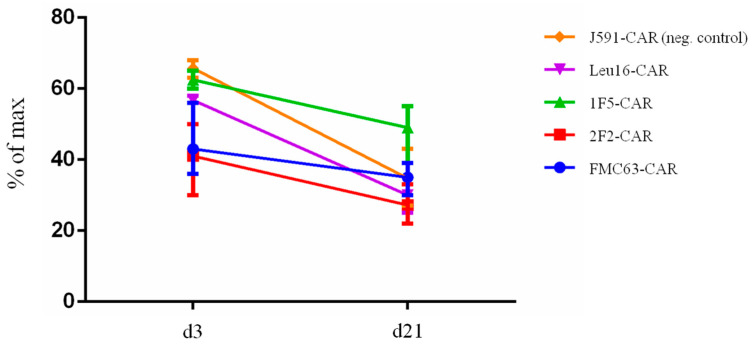
CAR T cell products display a similar decrease in the percentage of CAR-positive cells upon long-term cultivation (day 21 vs. day 3).

**Figure 6 ijms-24-03698-f006:**
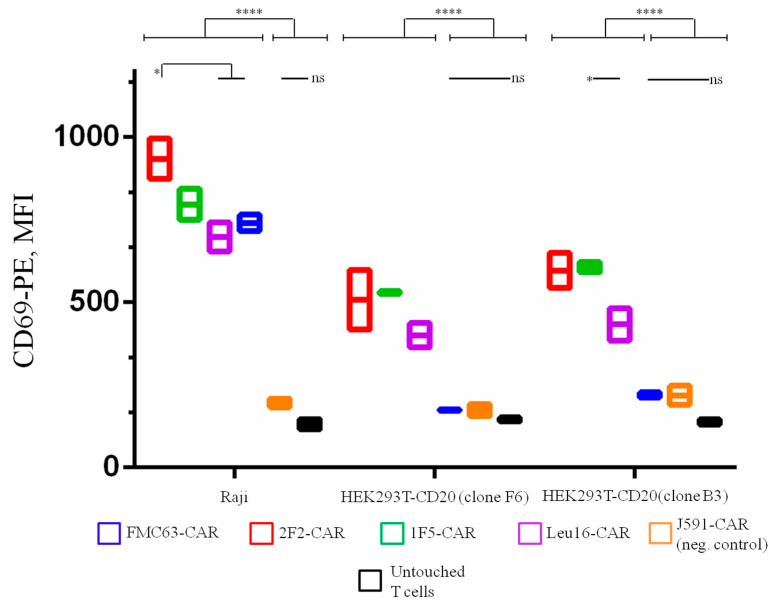
CD20-targeting CAR T cells display specific activation (CD69 expression) upon co-incubation with cell targets displaying high (Raji), moderate (clone F6), or low (clone B3) CD20 expression. Lower levels of CD69 expression are observed upon reduced CD20 surface density. Mean values with range are shown, * *p* < 0.05, **** *p* < 0.0001.

**Figure 7 ijms-24-03698-f007:**
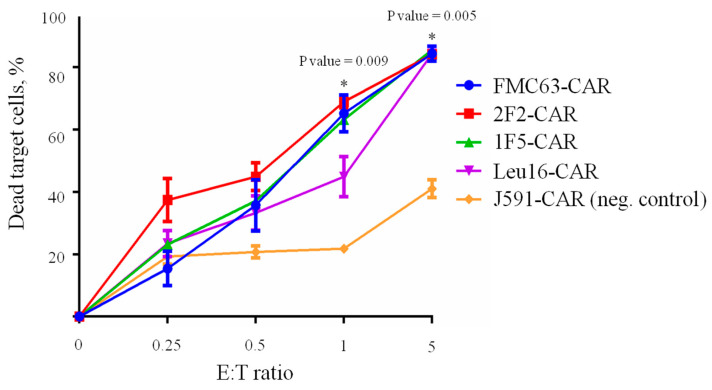
CD19- and CD20-specific CAR T cells (E) display similar cytotoxicity against target (T) Nalm6-CD20 cells (CD19+, CD20+) upon 4 h co-incubation across various E:T ratios. Irrelevant PSMA-specific J591-CAR T cells are used as the negative control. Mean values with range are shown, * *p* < 0.05.

**Figure 8 ijms-24-03698-f008:**
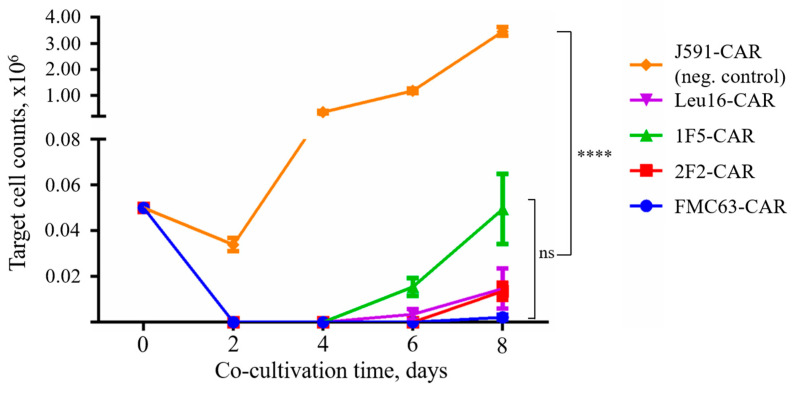
CAR T cells display pronounced sequential killing activity. Irrelevant PSMA-specific J591-CAR was used as a negative control (unrestrained target cell proliferation). Mean values with range are shown, **** *p* < 0.0001.

**Figure 9 ijms-24-03698-f009:**
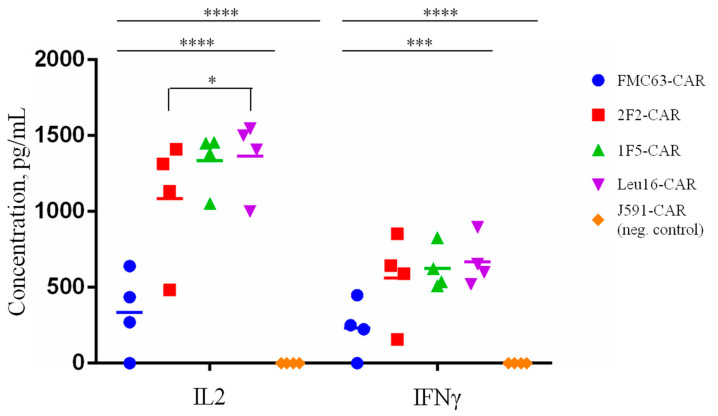
IL2 and IFNg secretion levels following CAR T cell co-incubation (E:T = 1:2) with Nalm6-CD20 target tumor cells. Average values of IL2 and IFNg concentrations in the supernatants of CAR T/target cell co-cultures are shown for each of the four donors (two technical replicates). Mean values with range are shown, * *p* < 0.05, *** *p* < 0.001, **** *p* < 0.0001.

**Figure 10 ijms-24-03698-f010:**
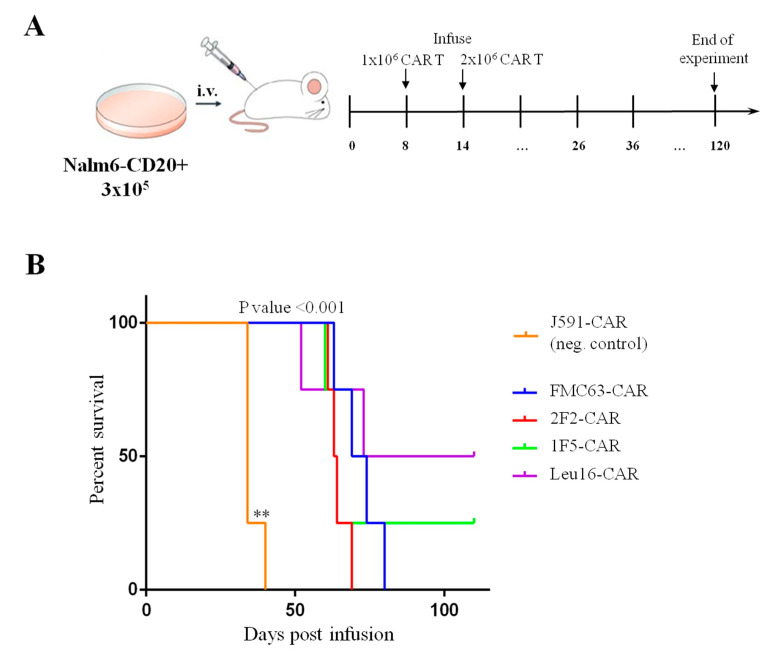
(**A**) Schematic illustration of the in vivo experimental design. (**B**) Infusion of 1F5- and Leu16-based CAR T cells results in longer survival of the xenotransplanted mice compared to that of FMC63- and 2F2-CAR T cells (Kaplan–Meier survival curves). All CD19- and CD20-targeted CAR T cells display significantly stronger in vivo functionality compared to the irrelevant PSMA-specific J591-CAR T cells. Differences between survival curves were analyzed by Log-rank (Mantel–Cox) test, ** *p* < 0.01.

**Figure 11 ijms-24-03698-f011:**
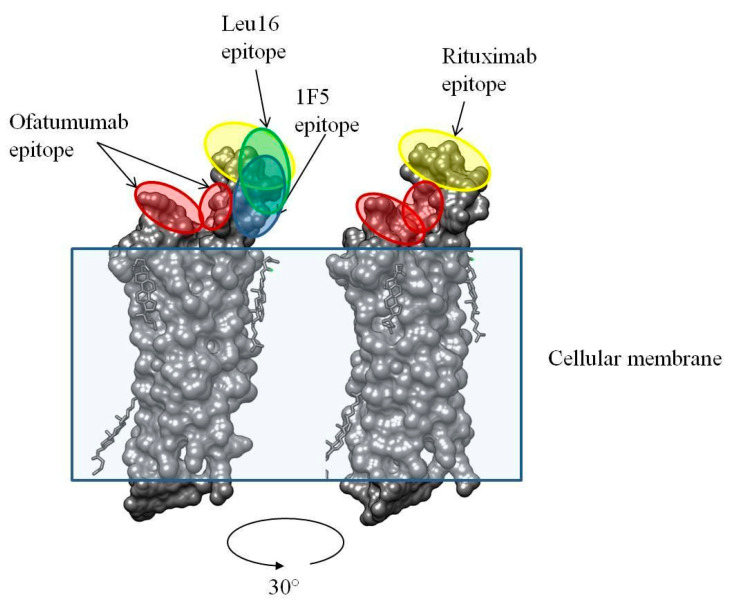
3D structure of human CD20 and the binding epitopes for ofatumumab and rituximab [52].

**Table 1 ijms-24-03698-t001:** Characteristics of CAR T cell products produced in evaluation runs.

	Fold Expansion	% CAR	% of Cytotoxicity, E:T 1:1
1. FMC63-CAR	26.1 ± 3.7	43 ± 4%	65 ± 5%
2. 2F2-CAR	29.7 ± 3.7	41 ± 4%	69 ± 6%
3. 1F5-CAR	30.2 ± 3.5	62 ± 1%	63 ± 2%
4. Leu16-CAR	28.6 ± 3.9	57 ± 1%	45 ± 3%
5. J591-CAR (neg. control)	27.1 ± 4.0	66 ± 1%	20 ± 1%

Fold expansion, % CAR expression, and % cytotoxicity at E:T = 1 were assayed on days 12, 3, and 14, respectively.

## Data Availability

All data generated or analyzed during this study are included in the manuscript and its Appendix A.

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
