# Peer review of "Comparative Pre-Clinical Analysis of CD20-Specific CAR T Cells Encompassing 1F5-, Leu16-, and 2F2-Based Antigen-Recognition Moieties"

_ijms, 2023, doi:10.3390/ijms24043698_

Round 1
Reviewer 1 Report
Overall intriguing study evaluating kinetics and cytotoxic potential over various CAR constructs employing murine based scFV vs humanized model. Overall a good study and relevant to the field. I have no concerns with the present work although would recommend that Figure 5 label the Y axes.
Author Response
Dear Reviewer,
Thank you for your valuable review.
Please find below our point-by-point responses
Figure 5 label the Y axes."
Appropriate changes to the y axis labels have been introduced.
Reviewer 2 Report
Belovezhets et al., constructed three different CD20-specific CAR T cells to target a leukemia cell line Nalm-6, and evaluated their in vitro and in vivo anti-tumor efficacy. Different anti-CD20 CAR constructs only differ in the antigen-recognition scFv, and the specificity against CD20 was designed based on mouse monoclonal antibodies 1F5 and Leu16, or human monoclonal antibody 2F2. Data showed that different CAR scFv constructs did not affect lentiviral transduction efficiency, and the subpopulation composition was comparable. However, different constructs resulted in distinct tonic signaling and sequential killing of Nalm-6 in vitro. In addition, Leu16-based anti-CD20 CAR T cells (BBζ) exhibited the strongest in vivo potency among different constructs. Overall, this study provides interesting results in terms of evaluating anti-CD20 CAR T cell efficacy with different scFv specificity using both in vitro and in vivo models, however, the current data do not provide enough support for their conclusions, and the comparison between different groups is not complete nor convincing. Here are the comments in detail:
1. Figure 1A, J591-derived scFv construct design was missing in the schematic illustration in 1A. According to the manuscript, it was made using the same backbone as other CARs. Please make this clear.
2. Figure 2, abbreviations such as TN, TCM, TEM, etc, should be defined where they are mentioned for the first time. What are the markers used to define these subpopulations? Interpretation of the subpopulation data based on TN, TCM, or TEM is not included in the manuscript. Please discuss the memory markers if the authors decide to show the results.
3. Figure 3, tonic signaling is an interesting feature of CAR-T cells to look at, however, the authors used “CAR T cell proliferation dynamics in the absence of target cell stimulation as an indirect measure of tonic signaling”. Please provide any relevant citations or references that support this decision. Please also consider conducting activation marker surface staining using flow cytometry, or molecular immunology assays such as Western blots to support the read-outs of tonic signaling comparisons.
4. Figure 5, please label the y-axis. Since the authors claimed that Leu16- and 1F5-based CAR T cells performed better in the sequential killing assays, please provide statistically significant in vitro data to support this conclusion. No biological replicates were mentioned, and please consider possible “batch effects” which might contribute to the current observed data.
5. Figure 6, please consider including a schematic illustration of the in vivo experimental design.
Overall, this study provided a good start in evaluating the efficacy of several newly designed anti-CD20 CAR T cell products. However, a major limitation is that only one donor was included in this study, reducing its reliability and generalizability. The in vivo experiment should be expanded and consider including multiple read-outs to further strengthen the statements.

Author Response
Dear Reviewer,
Thank you for your valuable review.
Please find below our point-by-point responses
We appreciate the reviewer’s thorough analysis and suggestions. In order to address the concerns raised and to adequately compare various aspects of CAR T cell activity, all CAR T cell experiments were redone, this time using the T cells from four donors, which raises our confidence in the validity of our conclusions.
1. Figure 1A, J591-derived scFv construct design was missing in the schematic illustration in 1A. According to the manuscript, it was made using the same backbone as other CARs. Please make this clear.
The figure has been corrected.
2. Figure 2, abbreviations such as TN, TCM, TEM, etc, should be defined where they are mentioned for the first time. What are the markers used to define these subpopulations? Interpretation of the subpopulation data based on TN, TCM, or TEM is not included in the manuscript. Please discuss the memory markers if the authors decide to show the results.
Appropriate changes have been introduced. Memory markers used in study are described.
3. Figure 3, tonic signaling is an interesting feature of CAR-T cells to look at, however, the authors used “CAR T cell proliferation dynamics in the absence of target cell stimulation as an indirect measure of tonic signaling”. Please provide any relevant citations or references that support this decision. Please also consider conducting activation marker surface staining using flow cytometry, or molecular immunology assays such as Western blots to support the read-outs of tonic signaling comparisons.
This part has been restructured to meet the reviewer’s suggestions. Namely, we now incorporate the data for the phenotypic composition of CAR T cell products after their prolonged re-stimulation with target cells.
4. Figure 5, please label the y-axis. Since the authors claimed that Leu16- and 1F5-based CAR T cells performed better in the sequential killing assays, please provide statistically significant in vitro data to support this conclusion. No biological replicates were mentioned, and please consider possible “batch effects” which might contribute to the current observed data.
The experiment was re-done anew using CAR T cell products obtained from four donors. Appropriate changes to the Figure have now been introduced.
5. Figure 6, please consider including a schematic illustration of the in vivo experimental design.
The outline of the experimental design is now part of this figure to gain clarity.
Overall, this study provided a good start in evaluating the efficacy of several newly designed anti-CD20 CAR T cell products. However, a major limitation is that only one donor was included in this study, reducing its reliability and generalizability. The in vivo experiment should be expanded and consider including multiple read-outs to further strengthen the statements."
We thank the reviewer for this comment. To address this issue, all CAR T cell experiments have been redone and this time they used the T cells obtained from four donors. In vivo experiment in an expanded format could not be repeated for financial and logistic reasons. To avoid overinterpreting the data and address the reviewer’s concern, the conclusions have been toned down.
Reviewer 3 Report
In this manuscript the authors compared several chimeric antigen receptor T (CAR-T) cells directed against the CD20 molecule. Anti-CD19 CAR-T cell therapy has greatly improved the treatment and prognosis of hematological malignancies. However, a significant proportion of patients relapse after treatment and CD19 tumor antigen loss has been described in several studies. Therefore, characterization and manufacturing of CAR-T cells targeting alternative tumor antigens is highly important. However, the authors failed to address central questions in the manuscript, e.g. affinity/avidity of the used CAR receptors, cytokine production, cell longevity. The presented data showed no significant difference in the cytotoxic abilities of the tested CAR-T cells. Moreover, it is unclear how often the analyses were performed and whether all measurements were done with CAR-T cells manufactured from only one healthy donor. The conclusions are often overstated and some data are not entirely believable (e.g. Supp. Fig. 1: 71% FMC63 vs. 63% 1F5 although the negative peak is higher in the former case).
In addition to these thematic difficulties, there are a number of other issues. The manuscript has to be corrected in grammar and syntax, space characters are often missing. Material and methods have to be improved (antibody clones, instruments, companies, … are missing). The statistical tests used were not stated, some results lack statistical analysis or it is unclear what was compared. The gating strategy is missing, as are some labels (y-axis Fig. 2 and 5, text Supp. Fig. 1). Discussion of the data should be excluded from the result section.
Author Response
Dear Reviewer,
Thank you for your valuable review.
Please find below our point-by-point responses
In this manuscript the authors compared several chimeric antigen receptor T (CAR-T) cells directed against the CD20 molecule. Anti-CD19 CAR-T cell therapy has greatly improved the treatment and prognosis of hematological malignancies. However, a significant proportion of patients relapse after treatment and CD19 tumor antigen loss has been described in several studies. Therefore, characterization and manufacturing of CAR-T cells targeting alternative tumor antigens is highly important. However, the authors failed to address central questions in the manuscript, e.g. affinity/avidity of the used CAR receptors, cytokine production, cell longevity.
These are excellent suggestions. To address these concerns, all CAR T cell experiments were re-done, this time using the starting material from four donors.
Unfortunately, there are only very limited published data on the affinity of the antigen-recognition modules used in our study. This is largely attributable to the complex structure of CD20 (it has several transmembrane regions and two extracellular loops that do not fold natively in the absence of the transmembrane regions), hence CD20 is not a straightforward target to be produced as a soluble recombinant protein and to be used for affinity measurements. It is for this reason that classical approaches such as SPR, BLI, and ITC are not applicable to measurements of CAR affinity to the full-length CD20.
Cytokine secretion has now been studied and the data are presented in the manuscript (Fig 7).
The presented data showed no significant difference in the cytotoxic abilities of the tested CAR-T cells. Moreover, it is unclear how often the analyses were performed and whether all measurements were done with CAR-T cells manufactured from only one healthy donor.
To address this concern, all experiments have been repeated using the material from four donors. Appropriate changes have been introduced into the text.
The conclusions are often overstated and some data are not entirely believable (e.g. Supp. Fig. 1: 71% FMC63 vs. 63% 1F5 although the negative peak is higher in the former case).
Supplement Fig 1 was definitely incorrect in the original submission, and a correct figure is now included.
In addition to these thematic difficulties, there are a number of other issues. The manuscript has to be corrected in grammar and syntax, space characters are often missing.
The manuscript is now professionally proofread by the native speaker.
Material and methods have to be improved (antibody clones, instruments, companies, … are missing).
Appropriate details have been incorporated into Materials and Methods.
The statistical tests used were not stated, some results lack statistical analysis or it is unclear what was compared. The gating strategy is missing, as are some labels (y-axis Fig. 2 and 5, text Supp. Fig. 1).
Presentation of the statistical tests and illustrations have been adjusted to address this concern.
Discussion of the data should be excluded from the result section.
Discussion of data is now appropriately excluded from the Results. Significant proportion of the text has been re-written.
Round 2
Reviewer 2 Report
Belovezhets et al. made a significant revision to the project which aims to construct different CD20-specific CAR T cells to target leukemia cell lines and to evaluate their in vitro and in vivo anti-tumor efficacy. Here is a minor suggestion:
1. Figure 8, the data point for the J591-CAR T cell on day 4 is missing. Please fix this or explain why it is missing.
Building upon the previous comments, the authors provide more in-depth and statistically significant results in their in vitro studies. Overall, this version provides a reasonable amount of information on evaluating the efficacy of several newly designed anti-CD20 CAR T cell products.
Author Response
Dear Reviewer,Thank you for your valuable comment:
"1. Figure 8, the data point for the J591-CAR T cell on day 4 is missing. Please fix this or explain why it is missing.".
The plot (Figure 8) is now corrected, the data point for the J591-CAR T cell on day 4 point was missing due to issues with the scaling of the axes.
Reviewer 3 Report
In this manuscript Belovezhets and colleagues compared the phenotype and activity of three different CD20-directed CAR T cells with anti-CD19 CAR T cells. The authors reworked the article and addressed some of my concerns. However, the originality and novelty of the manuscript is still limited and no valuable conclusions can be drawn from the analyses. All different CAR T cells showed similar differentiation, expansion and cytotoxic abilities. The authors haven’t included affinity/avidity of the different CAR T cells (e.g. signaling threshold and strength after stimulation with recombinant CD20 (coated to beads)), nor deep immune phenotyping after chronic stimulation (stimulatory/inhibitory receptors, cytotoxic molecules,…). The only differences between the CAR T cell-populations were (i) impaired effector-memory differentiation after chronic antigen stimulation and (ii) reduced cytokine production of CD19-directed CAR T cells – but these observations were not further investigated. In addition, the authors should include the gating strategy and raw data/dotplots of the T cell differentiation analyses. It’s hard to believe that 30-60% of CAR T cells have a naïve phenotype after 21 days of expansion. Finally the last sentence of the result section (“Hence,…) should be excluded as there are no (significant) differences in the therapeutic efficacy of CAR T cells.
Author Response
Dear Reviewer,
Thank you for your valuable comments, we believe that while addressing your questions we were able to improve the quality of our research and to provide a stricter discussion of the results. Please find our point-by-point responses below.
In this manuscript Belovezhets and colleagues compared the phenotype and activity of three different CD20-directed CAR T cells with anti-CD19 CAR T cells. The authors reworked the article and addressed some of my concerns. However, the originality and novelty of the manuscript is still limited and no valuable conclusions can be drawn from the analyses. All different CAR T cells showed similar differentiation, expansion and cytotoxic abilities. The authors haven’t included affinity/avidity of the different CAR T cells (e.g. signaling threshold and strength after stimulation with recombinant CD20 (coated to beads)), nor deep immune phenotyping after chronic stimulation (stimulatory/inhibitory receptors, cytotoxic molecules,…).
We included an experiment assessing CAR T cells ability to activate upon stimulation by the targets with low CD20 expression. Difference in CD69 expression among CD20-specific CAR T cells was observed and is discussed in the manuscript. Also, we performed evaluation of inhibitory receptor expression after repetitive stimulation of CAR T cells with targets.
The only differences between the CAR T cell-populations were (i) impaired effector-memory differentiation after chronic antigen stimulation and (ii) reduced cytokine production of CD19-directed CAR T cells – but these observations were not further investigated. In addition, the authors should include the gating strategy and raw data/dotplots of the T cell differentiation analyses.
Gating strategy and more flow data are now included in the supplement. Additional experiments highlighted the functional variability between CAR T cells of different design.
It’s hard to believe that 30-60% of CAR T cells have a naïve phenotype after 21 days of expansion. Finally the last sentence of the result section (“Hence,…) should be excluded as there are no (significant) differences in the therapeutic efficacy of CAR T cells.
Representative raw dotplots of CAR T cells staining were included in the supplement. The last part of the Results section was re-written.